# T-Projection: High Quality Annotation Projection
# for Sequence Labeling Tasks

**Iker García-Ferrero    Rodrigo Agerri    German Rigau**
HiTZ Center - Ixa, University of the Basque Country UPV/EHU
{ iker.garciaf, rodrigo.agerri, german.rigau }@ehu.eus

## Abstract

In the absence of readily available labeled data for a given sequence labeling task and language, annotation projection has been proposed as one of the possible strategies to automatically generate annotated data. Annotation projection has often been formulated as the task of transporting, on parallel corpora, the labels pertaining to a given span in the source language into its corresponding span in the target language. In this paper we present T-Projection, a novel approach for annotation projection that leverages large pretrained text-to-text language models and state-of-the-art machine translation technology. T-Projection decomposes the label projection task into two subtasks: (i) A candidate generation step, in which a set of projection candidates using a multilingual T5 model is generated and, (ii) a candidate selection step, in which the generated candidates are ranked based on translation probabilities. We conducted experiments on intrinsic and extrinsic tasks in 5 Indo-European and 8 low-resource African languages. We demostrate that T-projection outperforms previous annotation projection methods by a wide margin. We believe that T-Projection can help to automatically alleviate the lack of high-quality training data for sequence labeling tasks. Code and data are publicly available.[1]

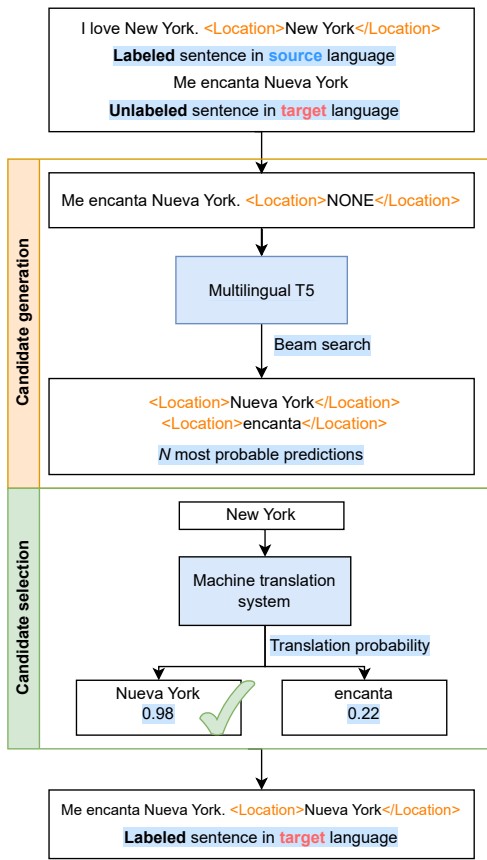

Figure 1: T-Projection two-step method to project sequence labeling annotations across languages.

## 1 Introduction

The performance of supervised machine-learning methods for Natural Language Processing, including advanced deep-neural models (Devlin et al., 2019; Conneau et al., 2020; Xue et al., 2021; Zhang et al., 2022), heavily depends on the availability of manually annotated training data. In addition, supervised models show a significant decrease in performance when evaluated in out-of-domain settings (Liu et al., 2021). This means that obtaining optimal results would require to manually generate annotated data for each application domain and language, an unfeasible task in terms of monetary cost and human effort. As a result, for the majority of languages in the world the amount of manually annotated data for many downstream tasks is simply nonexistent (Joshi et al., 2020).

The emergence of multilingual language models (Devlin et al., 2019; Conneau et al., 2020) allows for zero-shot cross-lingual transfer. A model fine-tuned on one language, typically English, can be directly applied to other target languages. However, better results can be obtained by either machine translating the training data from English into the

---

[1] 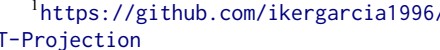 https://github.com/ikergarcia1996/T-Projection

target languages or, conversely, translating the test data from the target language into English (Hu et al., 2020; Artetxe et al., 2023).

Sequence labeling tasks, which involve span-level annotations, require an additional step called *annotation projection*. This step involves identifying, in the translated sentence, the sequence of words that corresponds to the labeled spans in the source text (Yarowsky et al., 2001; Ehrmann et al., 2011). The majority of previous published work on this line of research explores the application of word-alignments (Ehrmann et al., 2011). However, projection methods based on word-alignments have achieved mixed results as they often produce partial, incorrect or missing annotation projections (García-Ferrero et al., 2022). This is due to the fact that word alignments are computed on a word-by-word basis leveraging word co-occurrences or similarity between word vector representations. That is, without taking into consideration the labeled spans or categories to be projected. Other techniques have also been proposed, such as fine-tuning language models in the span projection task (Li et al., 2021), translating the labeled spans independently from the sentence (Zhou et al., 2022) or including markers during the machine translation step (Chen et al., 2023). However, automatic annotation projection remains an open and difficult challenge.

In this paper we present T-Projection, a novel approach to automatically project sequence labeling annotations across languages. Our method is illustrated by Figure 1. We split the annotation projection task into two steps. First, we use mT5 (Xue et al., 2021) text-to-text model to generate a set of projection candidates in the target sentence for each labeled category in the source sentence. This step exploits the labeled categories as well as the cross-lingual capabilities of large pretrained multilingual language models. Second, we rank the candidates based on the probability of being generated as a translation of the source spans. We use the M2M100 (Fan et al., 2021) and NLLB200 (Costa-jussà et al., 2022) state-of-the-art MT models to compute the translation probabilities (Vamvas and Sennrich, 2022).

We conduct an intrinsic evaluation on three different tasks, Opinion Target Extraction (OTE), Named Entity Recognition (NER) and Argument Mining (AM), and five different target languages (French, German, Italian, Russian and Spanish). In this evaluation we compare the label projections

generated by various systems with manually projected annotations. On average, **T-Projection improves the current state-of-the-art annotation projection methods by more than 8 points in F1 score**, which constitutes a significant leap in quality over previous label projection approaches. Additionally, we performed a real-world NER task evaluation involving eight low-resource African languages. In this downstream evaluation, T-Projection outperformed other annotation projection methods by 3.6 points in F1 score.

## 2 Background

While most of the previous approaches for annotation projection are based on the application of word alignments, other techniques have also been explored.

### 2.1 Word alignment methods

Ehrmann et al. (2011) use the statistical alignment of phrases to project the English labels of a multi-parallel corpus into the target languages. Instead of using discrete labels, Wang and Manning (2014) project model expectations with the aim of facilitating the transfer of model uncertainty across languages. Ni et al. (2017) aim to filter good-quality projection-labeled data from noisy data by proposing a set of heuristics. Other works have proposed to use word alignments generated by Giza++ (Och and Ney, 2003) to project parallel labeled data from multiple languages into a single target language (Agerri et al., 2018). Fei et al. (2020) use the word alignment probabilities calculated with FastAlign (Dyer et al., 2013) and the POS tag distributions of the source and target words to project from the source corpus into a target language machine-translated corpus. Finally, García-Ferrero et al. (2022) propose an annotation projection method based on machine translation and AWESOME (Dou and Neubig, 2021), Transformer-based word alignments to automatically project datasets from English to other languages.

### 2.2 Other projection methods

With respect to projection methods which do not use word alignments, Jain et al. (2019) first generate a list of projection candidates by orthographic and phonetic similarity. They choose the best matching candidate based on distributional statistics derived from the dataset. Xie et al. (2018) propose a method to find word translations based

on bilingual word embeddings. Li et al. (2021) use a XLM-RoBERTa model (Conneau et al., 2020) trained with the source labeled part of a parallel corpus to label the target part of the corpus. Then they train a new improved model with both labeled parts. Zhou et al. (2022) first replace the labeled sequences with a placeholder token in the source sentence. Second, they separately translate the sentence with the placeholders and the labeled spans into the target sentences. Finally, they replace the placeholders in the translated sentence with the translation of the labeled spans. Chen et al. (2023) jointly perform translation and projection by inserting special markers around the labeled spans in the source sentence. To improve the translation accuracy and reduce translation artifacts, they fine-tune the translation model with a synthetic label protection dataset.

To summarize, previous work does not take advantage of all the information which is available while performing annotation projection. For example, word alignment models do not take into account the labeled spans and their categories during alignment generation. Instead, they simply rely on information about word co-occurrences or similarity between word representations. Those techniques based on MT to generate the target part of the parallel corpus ignore additional knowledge that the MT model encodes. Furthermore, methods that utilize MT models for both translation and projection often introduce translation artifacts, which can affect the quality and accuracy of the projections. In contrast, our T-Projection method exploits both the labeled spans and their categories together with the translation probabilities to produce high-quality annotation projections.

## 3 T-Projection

Given a source sentence in which we have sequences of words labeled with a class, and its parallel sentence in a target language, we want to project the labels from the source into the target. As illustrated in Figure 1, T-Projection implements two main steps. First, a set of projection candidates in the target sentence are generated for each labeled sequence in the source sentence. Second, each projection candidate is ranked using a machine translation model. More specifically, candidates are scored based on the probability of being generated as a translation of the source labeled sequences. While the *candidate generation* step exploits the

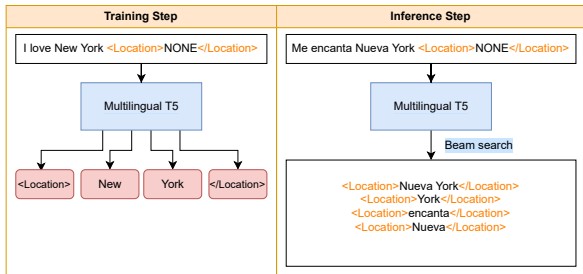

Figure 2: Illustration of the candidate generation step. For each label, we generate a set of probable candidates.

labeled spans and their categories in the source sentence as well as the zero-shot cross-lingual capabilities of large pretrained multilingual language models, the *candidate selection* step applies state-of-the-art MT technology to find those projection candidates that constitute the best translation for each source labeled span. As demonstrated by our empirical evaluation in Sections 5 and 6, we believe that these two techniques leverage both the available information and knowledge from the annotated text and language models thereby allowing us to obtain better annotation projections. These two steps are described in detail in the following two subsections.

### 3.1 Candidate Generation

When trying to project labeled sequences from some source data into its parallel target dataset, we would expect both the source and the target to contain the same number of sequences labeled with the same category. For example, given the English source sentence "*<Person>Obama</Person> went to <Location>New York</Location>*" and its parallel Spanish unlabeled target sentence "*Obama fue a Nueva York*", we would expect to find the same two entities (*person* and *location*) in the target sentence. To solve the task of candidate generation, we finetune the text-to-text mT5 (Xue et al., 2021) model using a HTML-tag-style prompt template (Huang et al., 2022). As illustrated by Figure 2, the input consists of concatenating the unlabeled sentence followed by a list of tags ("*<Category>None</Category>*") with the category of each labeled span that we expect to find in the sentence. If two or more spans share the same category then we append the tag as many times as spans are expected with that category.

Unlike Huang et al. (2022), we do not encode the tags for each category as special tokens. Instead, we verbalize the categories (i.e PER->Person) and

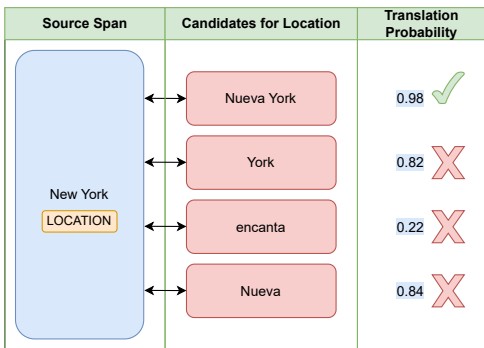

Figure 3: Candidate selection: candidates are scored based on the probability of being generated as a translation of the originally labeled sequences.

use the token representations already existing in the model. We expect that, thanks to the language modeling pretraining, T5 would have a good semantic representation of categories such as *Person*, *Location*, *Claim*, etc.

As Figure 2 illustrates, we finetune mT5 with the labeled source dataset. We train the model to replace the token *None* with the sequence of words in the input sentence that corresponds to that category. We use Cross-Entropy loss for training.

At inference, we label the target sentences which are parallel translations of the source sentences. As the source tells us how many labeled spans should we obtain in the target, we use the labels of the corresponding source sentence to build the prompts. In other words, our objective is to label parallel translations of the sentences we used for training. We take advantage of the zero-shot cross-lingual capabilities of mT5 to project the labels from the source to the target sentence. The output tokens are generated in an autoregressive manner. We use beam search decoding with 100 beams to generate 100 candidates for each input tag.

## 3.2 Candidate Selection

As depicted in Figure 3, all the generated candidates are grouped by category. In other words, if the previous step has generated multiple spans with the same category (i.e, two *locations* in a sentence) then all the candidates are included in a single set. Furthermore, all the candidates that are not a subsequence of the input sentence are filtered out.

For each labeled span in the source sentence, we rank all the projection candidates that share the same category as the source span using their translation probabilities (also known as translation equivalence) which have been obtained by apply-

ing the pretrained M2M100 (Fan et al., 2021) or NLLB200 (Costa-jussà et al., 2022) MT models and the *NMTScore* library[2] (Vamvas and Sennrich, 2022). Thus, given the source span **A** and the candidate **B** the translation probability is computed as follows (Vamvas and Sennrich, 2022):

$$p_{\theta_a}(A \mid B) := \left[ \prod_{i=0}^{|A|} p_{\theta_a}\left(A^i \mid B, A^{<i}\right) \right]^{\frac{1}{|A|}}$$

The translation probability is normalized:

$$sim(A \mid B) = \frac{p_{\theta_a}(A|B)}{p_{\theta_a}(A|A)}$$

As the translation probability varies depending on the direction of the translation, the scores are symmetrized by computing the scores of both translation directions and averaging them:

$$sim(A, B) = \frac{1}{2}sim(A \mid B) + \frac{1}{2}sim(B \mid A)$$

Finally, for each labeled span in the source sentence, we choose the candidate in the target with the highest translation probability. Once a candidate has been selected, that candidate, and any other that overlaps with it, is removed from the set of possible candidates. In this way we prevent assigning the same candidate in the target to two different spans in the source.

## 4 Experimental Setup

In order to evaluate our method we perform both intrinsic and extrinsic evaluations.

**Intrinsic evaluation:** We select a number of datasets that have been manually projected from English into different target languages. The manual annotations are used to evaluate and compare T-Projection with respect to previous state-of-the-art label projection models. Results are reported by computing the usual *F1-score* used for sequence labelling (Tjong Kim Sang, 2002) with the aim of evaluating the quality of the automatically generated projections with respect to the manual ones.

**Extrinsic evaluation:** In this evaluation we assess the capability of T-Projection to automatically generate training data for sequence labeling tasks, NER in this particular case. The process begins by utilizing the machine translation system NLLB200 (Costa-jussà et al., 2022) to translate data from English into 8 low-resource African languages. We then project the labels from English onto the respective target languages. The automatically generated datasets are then employed to train NER models,

---

[2]https://github.com/ZurichNLP/nmtscore

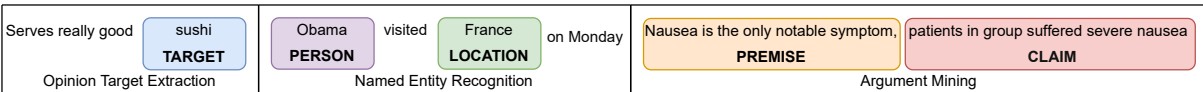

Figure 4: Sequence labeling tasks in our experiments

which are evaluated using a relatively small manually annotated test set. The same procedure is performed with other state-of-the-art label projection models. The comparison of the results obtained is reported in terms of *F1 score*.

## 4.1 Datasets

The datasets used correspond to three sequence labeling tasks which are illustrated by Figure 4.

**Opinion Target Extraction (OTE)** Given a review, the task is to detect the linguistic expression used to refer to the reviewed entity. We use the English SemEval 2016 Aspect Based Sentiment Analysis (ABSA) datasets (Pontiki et al., 2014). Additionally, for the evaluation we also used the parallel versions for Spanish, French and Russian generated via machine translation and manual projection of the labels (García-Ferrero et al., 2022).

**Named Entity Recognition (NER)** The NER task consists of detecting named entities and classifying them according to some pre-defined categories. We use an English, Spanish, German, and Italian parallel NER dataset (Agerri et al., 2018) based on Europarl data (Koehn, 2005). Manual annotation for the 4 languages was provided following the CoNLL 2003 (Tjong Kim Sang, 2002) guidelines. In the extrinsic evaluation, we use MasakhaNER 2.0 (Adelani et al., 2022), a human-annotated NER dataset for 20 African languages.

**Argument Mining (AM)** In the AbstRCT English dataset two types of argument components, Claims and Premises, are annotated on medical and scientific texts collected from the MEDLINE database (Mayer et al., 2020). For evaluation we used its Spanish parallel counterpart which was generated following an adapted version of the method described above for OTE (Yeginbergenova and Agerri, 2023). In contrast to NER and OTE, the sequences in the AM task consist of very long spans of words, often encompassing full sentences. We use the Neoplasm split.

## 4.2 Baselines

We experiment with 4 different *word alignment systems*. Two statistical systems, **Giza++** (Och and Ney, 2003) and **FastAlign** (Dyer et al., 2013),

widely used in the field. We also evaluate two Transformer-based systems, **SimAlign** (Jalili Sabet et al., 2020) and **AWESOME** (Dou and Neubig, 2021), which leverage pre-trained multilingual language models to generate alignments. As the authors recommend, we use multilingual BERT (mBERT) (Devlin et al., 2019) as the backbone. We tested different models as backbone with no improvement in performance (see Appendix C). We use these four systems to compute word alignments between the source and the target sentences and generate the label projections applying the algorithm published by García-Ferrero et al. (2022)[3].

We also experiment with **EasyProject** (Chen et al., 2023), a system that jointly performs translation and projection by inserting special markers around the labeled spans in the source sentence. As this model generates its own translations it is therefore not suitable for the intrinsic evaluation which is why we only used it for the extrinsic evaluation.

We implemented two additional baselines for comparison. In the first baseline, inspired by Li et al. (2021), we use **XLM-RoBERTa** (Conneau et al., 2020) 3 billion parameter model (same parameter count as the mT5 model that we use in T-Projection) with a token classification layer (linear layer) on top of each token representation. We train the model in the source labeled dataset and we predict the entities in the translated target sentences. The second baseline adopts a **span translation** approach inspired by Zhou et al. (2022). We translate the labeled spans in the source sentence using the pretrained M2M100 12 billion parameter model and we match them with the corresponding span in the target sentence. For example, given the labeled source sentence "*I love <Location> New York </Location>.*" and the target sentence *"Me encanta Nueva York"*, we translate the span *New York* into the target language, resulting in the translated span *Nueva York* which is then matched in the target sentence. We employ beam search to generate 100 possible translations, and we select the most probable one that matches the target sentence.

---

[3]https://github.com/ikergarcia1996/Easy-Label-Projection

## 4.3 Models Setup

We use the 3 billion parameters pretrained mT5 (Xue et al., 2021) for the *candidate generation* step while *candidates are selected* using the M2M100 12 billion parameter machine translation model (Fan et al., 2021). In the case of MasakhaNER, since not all languages are included in M2M100, we resorted to NLLB200 (Costa-jussà et al., 2022) 3 billion parameter model instead, which was also used by the EasyProject method (Chen et al., 2023). Both MT models demonstrate comparable performance. For detailed hyperparameter settings, performance comparison of T-Projection using models of different sizes, and a comparison between T-Projection performance using M2M100 and NLLB200, please refer to the Appendix.

## 5 Intrinsic Evaluation

In this section we present a set of experiments to evaluate T-Projection with respect to current state-of-the-art approaches for annotation projection. We also analyze separately the performance of the *candidate generation* and *candidate selection* steps.

For the OTE task we train T-Projection and XLM-RoBERTa with the English ABSA 2016 training set. We also train the four word alignment systems (excluding SimAlign which is an unsupervised method) using the English training set together with the respective translations as parallel corpora. We augment the parallel data with 50,000 random parallel sentences from ParaCrawl v8 (Esplà et al., 2019). Models are evaluated with respect to the manually label projections generated by García-Ferrero et al. (2022). As the Europarl-based NER dataset (Agerri et al., 2018) provides only test data for each language, T-Projection and XLM-RoBERTa are trained using the full English CoNLL 2003 dataset (Tjong Kim Sang, 2002) together with the labeled English Europarl test data. The word alignment models are in turn trained with the the parallel sentences from the Europarl-based NER data plus 50,000 parallel sentences extracted from Europarl v8 (Koehn, 2005). We evaluate the model with respect to the manual annotations provided by Agerri et al. (2018). With respect to Argument Mining, we use the Neoplasm training set from the AbstRCT dataset to train T-Projection and XLM-RoBERTa, adding its Spanish translation as parallel corpora for the word alignment systems. As this is a medical text corpus, the parallel corpora is complemented with 50,000 parallel sentences from the WMT19 Biomedical Translation Task (Bawden et al., 2019). We evaluate the models with respect to the manually projected labels by Yeginbergenova and Agerri (2023).

## 5.1 Annotation Projection Quality

Table 1 reports the results of the automatically projected datasets generated by each projection method with respect to the human-projected versions of those datasets. The systems based on word alignments obtain good results across the board, especially those using language models, namely, SimAlign and AWESOME. In particular, AWESOME achieves good results for OTE and NER but very poor in AM. Manual inspection of the projections found out that AWESOME struggles to align articles and prepositions which are included in long sequences.

XLM-RoBERTa-xl shows a strong zero-shot cross-lingual performance. However, the generated datasets are of lower quality than the ones generated by the word-alignment systems. The results of the Span Translation approach are quite disappointing, especially when dealing with the long sequences of the AM task. Translating the labeled spans independently produce translations that, in many cases, cannot be located in the target sentence.

Our T-Projection method achieves the best results for every task and language. In OTE, it outperforms every other method by more than 2 points in F1 score averaged across the three languages. This suggests that T-Projection robustly projects labeled spans into machine-translated data. The NER evaluation is slightly different because the parallel data was translated by human experts. In this setting, T-Projection clearly improves AWESOME's results by 4.7 points, which constitutes a significant leap in the quality of the generated datasets.

Despite the fact that the word alignment systems have been trained using Europarl domain-specific data, and that most of the training data used for T-Projection is coming from the CoNLL-2003 dataset (news domain) plus very few annotated sentences (699) from Europarl, T-Projection still clearly obtains the best results in NER label projection. This suggests that our system can also be applied in out-of-domain settings.

Finally, T-Projection obtains the overall highest scores for Argument Mining which means that

|  | OTE | | | NER | | | AM | Avg |
|---|---|---|---|---|---|---|---|---|
|  | ES | FR | RU | ES | DE | IT | ES | |
| Giza++ (Och and Ney, 2003) | 77.0 | 73.3 | 72.4 | 73.3 | 75.3 | 68.4 | 86.6 | 77.7 |
| FastAlign (Dyer et al., 2013) | 75.0 | 72.9 | 76.9 | 70.2 | 77.0 | 67.0 | 85.7 | 77.4 |
| SimAlign (Jalili Sabet et al., 2020) | 86.7 | 86.3 | 87.7 | 85.4 | 87.4 | 81.3 | 84.1 | 85.3 |
| AWESOME (Dou and Neubig, 2021) | 91.5 | 91.1 | 93.7 | 87.3 | 90.7 | 83.1 | 54.8 | 78.0 |
| XLM-RoBERTa-xl (Conneau et al., 2020) | 80.2 | 76.2 | 74.5 | 73.9 | 68.3 | 73.9 | 66.5 | 71.8 |
| Span Translation | 66.5 | 46.3 | 58.7 | 68.8 | 63.5 | 69.2 | 21.6 | 48.7 |
| T-Projection | **95.1** | **92.3** | **95.0** | **93.6** | **94.0** | **87.2** | **96.0** | **93.9** |

Table 1: F1 scores for annotation projection in the OTE, NER and Argument Mining tasks.

our approach is particularly good projecting long sequences. Thus, T-Projection outperforms the second best model by 9.4 points in F1 score. In fact, the 96.0 F1 result obtained indicates that T-Projection is almost correctly projecting all the examples in the dataset.

If we look at the average over the three tasks and 5 languages, T-Projection improves by 8.6 points in F1 score the results of the second-best system, SimAlign. These results constitute a huge improvement over all the previous annotation projection approaches. To the best of our knowledge, these are by a wide margin the best annotation projection results published for sequence labeling.

### 5.2 The Role of the Candidates

|  | OTE | | | NER | | | AM | Avg |
|---|---|---|---|---|---|---|---|---|
|  | ES | FR | RU | ES | DE | IT | ES | |
| T-Projection | 95.1 | 92.3 | 95.0 | 93.6 | 94.0 | 87.2 | 96.0 | 93.9 |
| Ngrams + Candidate Selection | 89.7 | 86.1 | 93.8 | 83.8 | 79.3 | 73.3 | 73.5 | 80.7 |
| mT5 + Most Probable Candidate | 83.7 | 87.2 | 85.3 | 79.5 | 82.8 | 72.3 | 90.9 | 84.8 |
| mT5 + Upper bound | 98.6 | 97.0 | 97.9 | 98.0 | 98.5 | 94.0 | 99.3 | 98.0 |

Table 2: F1 scores for different candidate generation and candidate selection methods.

We perform a set of experiments to measure the relevance and performance of the *candidate generation* and *candidate selection* tasks. First, we replace mT5 with an ngram-based candidate generation approach. We consider as candidate spans every possible ngram with size 1..*sentence_length* (i.e *"Serves", "really", "good", "sushi", "Serves really"*...). Table 2 shows that this approach results in lower performance compared with our technique using mT5. Ngrams are much noisier than the candidates generated by mT5, most of them are

very similar to each other, and this makes selecting the right candidate a more challenging task. Thus, this experiment proves that our mT5 candidate generation approach is crucial to obtain good performance.

We also replace the *candidate selection* method with the *most probable candidate*. In other words, we only use the most probable beam generated by mT5 to label the target sentence. When using mT5 by itself, it obtains competitive results, close to those of the word alignment systems in Table 1. Still, the average performance drops by 9.2 points. This further confirms that both the *candidate generation* and *selection* steps are crucial for the T-Projection method.

In a final experiment we define an upperbound for *candidate selection* consisting of assuming that our model will always select the correct projection contained among the generated candidates. The upper bound achieves an average F1 score of 98. This result confirms with a very high probability that the correct candidate is almost always among the 100 candidates generated by mT5.

## 6 Extrinsic Evaluation

In this section we evaluate T-projection in a real-world low-resource scenario, namely, Named Entity Recognition in African Languages. We compare the results obtained by training on NER dataset automatically generated by T-Projection with respect to those automatically projected using two state-of-the-art label projection systems: AWESOME (The second-best NER system in Table 1) and EasyProject. We use the exactly same settings as Chen et al. (2023). For each target language in MasakhaNER2.0, we first translate the English CoNLL dataset using the NLLB-200 3 billion parameter model. Next, we project the English labels into the target language. It should be noted that EasyProject performs both of these processes in a

| Language | No. of Speakers | Lang family | Fine-tune$_{en}$ | AWESOME+EN | EasyProject+EN | T-Projection | T-Projection+EN |
|---|---|---|---|---|---|---|---|
| Hausa | 63M | Afro-Asiatic /Chadic | 71.7 | **72.7** | 72.2 | **72.7** | 72.0 |
| Igbo | 27M | NC / Volta-Niger | 59.3 | 63.5 | 65.6 | 71.4 | **71.6** |
| Chichewa | 14M | English-Creole | **79.5** | 75.1 | 75.3 | 77.2 | 77.8 |
| chiShona | 12M | NC / Bantu | 35.2 | 69.5 | 55.9 | **74.9** | 74.3 |
| Kiswahili | 98M | NC / Bantu | **87.7** | 82.4 | 83.6 | 84.5 | 84.1 |
| isiXhosa | 9M | NC / Bantu | 24.0 | 61.7 | 71.1 | **72.3** | 71.7 |
| Yoruba | 42M | NC / Volta-Niger | 36.0 | 38.1 | 36.8 | **42.7** | 42.1 |
| isiZulu | 27M | NC / Bantu | 43.9 | 68.9 | **73.0** | 66.7 | 64.9 |
| AVG | | | 54.7 | 66.5 | 66.7 | **70.3** | 69.8 |

Table 3: F1 scores on MasakhaNER2.0 for mDebertaV3 trained with projected annotations from different systems. "+EN" denotes concatenation of the automatically generated target language dataset with the source English dataset.

single step. Subsequently, we train an mDebertaV3 (He et al., 2021) model using the automatically generated datasets for each target language. Finally, this model is evaluated in the gold MasakhaNER2.0 test data. We only evaluate the 8 languages in MasakhaNER2.0 supported by mT5. We focus on named entities referring to Person, Location and Organization.

Table 3 presents the results of the evaluated models on the gold MasakhaNER2.0 test sets. For T-projection, we present the results of training with the automatically generated data for the target language only, and also by adding the original English CoNLL data concatenated with the automatically generated data for each target language. Regarding other systems, we only show the former results, as it was the only metric reported by previous work. In order to train and evaluate the NER models we apply the same hyperparameter settings and code as the authors of EasyProject.

The results show that T-projection achieves superior performance for seven out of the eight languages. Our model demonstrates a more pronounced performance difference in agglutinative languages such as Igbo and Shona. As outlined in Section 5, our model produces superior alignments compared to AWESOME. On the other side, we found that EasyProject, which utilizes markers for simultaneous translation and projection, introduces translation artifacts that hinder the performance of the downstream model. These artifacts are particularly noticeable in agglutinative languages, as EasyProject tends to separate words. For instance, in the case of Shona, consider the English sentence *"[Germany]'s representative to the [European Union]'s veterinary committee [Werner Zwingmann]"*. Our system produces the Shona sentence *"Mumiriri [weGermany] kukomiti yemhuka [yeEuropean Union] [Werner Zwingmann]"*, while EasyProject produces *"Mumiriri we [Germany]*

*ku [European Union] komiti yezvokurapa mhuka [Werner Zwingmann]"*. When training mDebertaV3 with T-projection generated data, which preserves the agglutinated words, we achieve better results compared to EasyProject that introduce artifacts by separating agglutinated words during translation and projection. Our system is only inferior in the Zulu language; however, on average, we improve the results by 3.6 F1 points. In contrast with previous work, our experiments revealed that concatenating English and translated data did not yield better results, potentially due to the superior quality of the data generated by T-Projection.

To the best of our knowledge, these are the best zero-shot results achieved for MasakhaNER2.0, underscoring the significant benefits of T-projection for NLP tasks in low-resource languages.

## 7 Concluding Remarks

In this paper we introduce T-Projection, a new annotation projection method that leverages large multilingual pretrained text-to-text language models and state-of-the-art machine translation systems. We conducted experiments on intrinsic and extrinsic tasks in 5 Indo-European languages and 8 African languages. T-projection clearly outperforms previous annotation projection approaches, obtaining a new state-of-the-art result for this task.

A comprehensive analysis shows that both the generation candidate and the candidate selection steps crucially contribute to the final performance of T-Projection. Future work includes adding more tasks and languages, especially those with different segmentation such as Japanese or Chinese. Unlike word alignment systems, T-Projection does not need to segment the words to do the projection which is why we think that our model can also be competitive to project annotations for many language pairs.

## Limitations

We evaluate the performance of T-Projection to project labels in sequence labeling tasks from English into a set of 5 Indo-European languages and 8 African languages. It would be interesting to evaluate the performance for other language families, which we leave for future work. Our model requires training a 3B parameter mT5 model. While training a 3B model is computationally expensive and requires a GPU with at least 48GB of VRAM, automatically generating a dataset is a one-off endeavour which results in a dataset usable for many occasions and applications, and much cheaper than manual annotation. Furthermore, we believe that the huge gains obtained by T-Projection justify the computation requirements. In any case, we expect that, thanks to the rapid development of computer hardware, the cost of T-Projection will be reduced in the near future. From a software perspective, recent advancements like 4-bit / 8-bit quantization (Dettmers et al., 2022b,a; Frantar et al., 2022) and Low Rank Adaptation (Hu et al., 2022) have the potential to reduce the hardware requirements of T-Projection.

## Acknowledgments

This work has been partially supported by the HiTZ center and the Basque Government (Research group funding IT-1805-22). We also acknowledge the funding from the following projects: (i) DeepKnowledge (PID2021-127777OB-C21) MCIN/AEI/10.13039/501100011033 and ERDF A way of making Europe; (ii) Disargue (TED2021-130810B-C21), MCIN/AEI/10.13039/501100011033 and European Union NextGenerationEU/PRTR (iii) Antidote (PCI2020-120717-2), MCIN/AEI/10.13039/501100011033 and by European Union NextGenerationEU/PRTR; (iv) DeepR3 (TED2021-130295B-C31) by MCIN/AEI/10.13039/501100011033 and EU NextGeneration programme EU/PRTR. Rodrigo Agerri currently holds the RYC-2017-23647 fellowship (MCIN/AEI/10.13039/501100011033 and by ESF Investing in your future). Iker García-Ferrero is supported by a doctoral grant from the Basque Government (PRE_2022_2_0208).

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

## A How many candidates do we need?

Generating candidates is expensive. The number of flops and memory usage increases linearly with the number of beams computed. Generating 20 candidates is twice as expensive as generating 10 candidates. We also need to add the extra workload of computing the similarity between more candidates. Figure 5 shows the average F1 score for each task when generating a different number of candidates. For OTE and NER small improvements are obtained when generating more than 25 candidates. However, in Argument Mining using a large number of candidates hurts T-Projection's performance, which performs optimally with just 10 candidates. While the results reported have been obtained generating 100 candidates, which is computationally very expensive, this analysis shows that we can use a much lower number of candidates and still achieve similar or even better results.

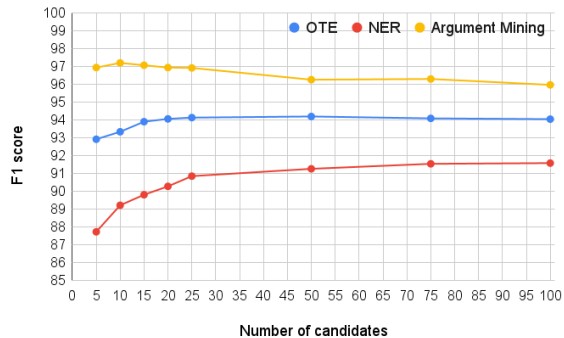

Figure 5: F1 score when generating a different number of candidates.

## B Model size vs Performance

|  | Model | #Params | OTE | NER | AM | Average |
|---|---|---|---|---|---|---|
| MT Size | m2m100 | 418M | 92.3 | 91.7 | 95.5 | 93.1 |
|  | m2m100 | 1.2B | 94.0 | **92.0** | 95.8 | **93.9** |
|  | m2m100 | 12B | **94.1** | 91.6 | 96.0 | **93.9** |
| mT5 size | mT5-small | 60M | 36.4 | 66.3 | 00.0 | 34.2 |
|  | mT5-base | 220M | 72.8 | 86.2 | 33.6 | 64.2 |
|  | mT5-large | 738M | 90.9 | 90.1 | 65.3 | 82.1 |
|  | mT5-xl | 3B | **94.1** | 91.6 | **96.0** | **93.9** |

Table 4: F1 scores of T-Projection when using translation and mT5 models of different size

We analyze the performance of T-Projection when using an mT5 model and a translation system with different number of parameters. Table 4 shows the average F1 performance across all the tasks and languages. First, we experiment with M2M100 models of different sizes. The results show that the size of the translation model does not have a significant impact on the performance of T-Projection.

However, the size of the mT5 model used does have a big impact on the final performance of the system. Although for OTE and NER switching from a 3B to a 738M parameter mT5 model produces competitive results, this is not the case when applied to AM. The overall trend is that when decreasing the number of parameters results keep decreasing. Summarizing, in order to achieve competitive performance for every task T-Projection requires a mT5 model with 3B parameters, although a 738M parameter model is still competitive for OTE and NER.

## C Tunning the Word Alignment Systems

To validate our results and further demonstrate the performance of T-Projection, we conduct a set of experiments that evaluate the performance of word-alignment systems under different settings. We first compare the annotation projection performance when using and not using 50,000 parallel sentences as data augmentation for training the word aligners. Note that in Section 5 all the results we show correspond to using 50,000 extra parallel sentences for training the word-alignment systems. As Table 5 shows, using the augmented dataset achieves the best performance. SimAlign (Dou and Neubig, 2021) and AWESOME (Dou and Neubig, 2021) recommend using their systems with multilingual-bert-cased (Devlin et al., 2019) as backbone. However, we also test XLM-RoBERTa-xl (Conneau et al., 2020) 3 billion parameter model with SimAlign and XLM-RoBERTa-large (355M parameters) model with AWESOME (The released AWESOME code at the time of writing this paper doesn't support XLM-RoBERTa-xl). Using XLM-RoBERTa produce worse results than using mBERT. These experiments show that we are using the word-alignment systems in their best-performing settings.

## D MT models vs Laser

We conducted experiments using M2M100-12B (Fan et al., 2021), NLLB200-3B (Costa-jussà et al., 2022) and prism (Thompson and Post, 2020) as model for computing translation probabilities. We also experiment with using LASER 2.0 (Artetxe and Schwenk, 2019) sentence representations in-

| System | Data Augmentation | Backbone | OTE | | | NER | | | AM | Average |
|---|---|---|---|---|---|---|---|---|---|---|
| | | | ES | FR | RU | ES | DE | IT | ES | |
| Giza++ (Och and Ney, 2003) | 0 | mBERT | 76.2 | 73.8 | 78.2 | 71.4 | 66.6 | 65.7 | 86.4 | 76.8 |
| FastAlign (Dyer et al., 2013) | 0 | mBERT | 72.3 | 70.4 | 74.6 | 60.3 | 64.0 | 57.5 | 84.0 | 72.4 |
| SimAlign (Jalili Sabet et al., 2020) | - | mBERT | 86.7 | 86.3 | 87.7 | 85.4 | 87.4 | 81.3 | 84.1 | 85.3 |
| AWESOME (Dou and Neubig, 2021) | 0 | mBERT | 88.9 | 89.8 | 91.2 | 86.1 | 89.4 | 83.0 | 57.1 | 77.8 |
| Giza++ (Och and Ney, 2003) | 50000 | mBERT | 77.0 | 73.3 | 72.4 | 73.3 | 75.3 | 68.4 | 86.6 | 77.7 |
| FastAlign (Dyer et al., 2013) | 50000 | mBERT | 75.0 | 72.9 | 76.9 | 70.2 | 77.0 | 67.0 | 85.7 | 77.4 |
| AWESOME (Dou and Neubig, 2021) | 50000 | mBERT | 91.5 | 91.1 | 93.7 | 87.3 | 90.7 | 83.1 | 54.8 | 78.0 |
| SimAlign (Jalili Sabet et al., 2020) | - | XLM-RoBERTa-xl | 86.2 | 86.1 | 89.5 | 85.8 | 88.4 | 81.2 | 76.9 | 83.1 |
| AWESOME (Dou and Neubig, 2021) | 50000 | XLM-RoBERTa-large | 86.1 | 86.1 | 87.4 | 87.2 | 87.5 | 83.1 | 54.8 | 75.8 |
| T-Projection | - | - | **95.1** | **92.3** | **95.0** | **93.6** | **94.0** | **87.2** | **96.0** | **93.9** |

Table 5: Results of the different word alignment systems when we train with and without a data augmentation corpus and different backbone models

| | OTE | | | NER | | | AM | Average |
|---|---|---|---|---|---|---|---|---|
| Candidate Scorer | ES | FR | RU | ES | DE | IT | ES | |
| Prism-745M | 91.4 | 86.8 | 94.3 | **93.8** | 93.4 | 85.4 | **96.3** | 92.7 |
| M2M100-12B | 95.1 | **92.3** | 95.0 | 93.6 | 94.0 | 87.2 | 96.0 | **93.9** |
| NLLB200-3B | **96.6** | 90.5 | **95.6** | 91.0 | **94.3** | **87.7** | 93.9 | 93.0 |
| LASER 2.0 | 89.0 | 80.6 | 91.3 | 91.2 | 91.6 | 86.5 | 70.4 | 82.4 |

Table 6: Results of T-Projection when selecting candidates using translation probability scores with different MT systems vs using the cosine similarity of the multilingual vector representations of the candidates computed using LASER 2.0

stead of the translation probabilities of NMTscore. We encode the source span as well as all the projection candidates using LASER encoder. We then rank them using cosine similarity. Table 6 shows the results. LASER2.0 is competitive when dealing with the short labeled sequences in the OTE and NER task. But the performance decreases when dealing with large sequences in the AM task. M2M100, NLLB200, and Prism exhibit comparable performance, with some of them achieving the best results in specific languages, but overall, their average performance is very similar.

## E Training details

We train the HuggingFace's (Wolf et al., 2019) implementation of mT5 [4] (3 billion parameter model) in the candidate generation step using the following hyper-parameters: Batch size of 8, 0.0001 learning rate, 256 tokens sequence length, cosine scheduler with 500 warn up steps and no weight decay. We use AdaFactor (Shazeer and Stern, 2018) optimizer. We train the model for 10 epochs in the OTE task, and 4 epochs for the NER and AM tasks. In the candidate selection step, we also use HuggingFace's

implementation of M2M100, and we use m2m100-12B-last-ckpt [5] checkpoint of M2M100 released by the authors. We use the direct-translation function of the NMTscore library to compute the translation probabilities. For MasakhaNER2.0 we use the training script and evaluation script developed by the authors [6] and the same hyper-parameter setup than Chen et al. (2023).

## F Dataset details

We list the size (number of sentences) of the dataset we use in Table 7. Note that all the datasets we use are parallel in all the languages, and the number of sentences is the same for all the languages.

---

[4] https://huggingface.co/google/mt5-xl

[5] https://huggingface.co/facebook/m2m100-12B-last-ckpt

[6] https://github.com/masakhane-io/masakhane-ner/blob/main/MasakhaNER2.0/scripts/mdeberta.sh

| Task | Split | Sentence No |
|---|---|---|
| **ABSA** | | |
| ABSA (Pontiki et al., 2014) | Train | 2000 |
| ABSA (Pontiki et al., 2014) | Test | 676 |
| **NER** | | |
| Europarl (Agerri et al., 2018) | Test | 799 |
| CoNLL03 (Tjong Kim Sang, 2002) | Train | 14987 |
| CoNLL03 (Tjong Kim Sang, 2002) | Dev | 3466 |
| CoNLL03 (Tjong Kim Sang, 2002) | Test | 3684 |
| MasakhaNER2.0 (Adelani et al., 2022) | Test (hau) | 1632 |
| MasakhaNER2.0 (Adelani et al., 2022) | Test (ibo) | 2180 |
| MasakhaNER2.0 (Adelani et al., 2022) | Test (sna) | 1772 |
| MasakhaNER2.0 (Adelani et al., 2022) | Test (swa) | 1882 |
| MasakhaNER2.0 (Adelani et al., 2022) | Test (xho) | 1632 |
| MasakhaNER2.0 (Adelani et al., 2022) | Test (yor) | 1963 |
| MasakhaNER2.0 (Adelani et al., 2022) | Test (nya) | 1784 |
| MasakhaNER2.0 (Adelani et al., 2022) | Test (zul) | 1669 |
| **AM** | | |
| AbsRCT Neoplasm (Mayer et al., 2020) | Train | 4404 |
| AbsRCT Neoplasm (Mayer et al., 2020) | Dev | 679 |
| AbsRCT Neoplasm (Mayer et al., 2020) | Test | 1251 |

Table 7: Size (Number of sentences) of the dataset we use to train and evaluate our systems.

For OTE, we use the SemEval-2016 Task 5 Aspect Based Sentiment Analysis (ABSA) dataset (Pontiki et al., 2014). We train T-Projection with the concatenation of the English train and test splits. We evaluate all the systems by projecting the training split. For NER we use the English, Spanish, German, Italian europarl parallel dataset from (Agerri et al., 2018). We train T-Projection with the concatenation of the English europarl dataset with the train, dev and test splits of the English CoNLL 2003 dataset (Tjong Kim Sang, 2002). We evaluate the systems by projecting the English NER europarl test splits. For Argument Mining, we use the AbstRCT Neoplasm English dataset (Mayer et al., 2020) and the Spanish AbsRCT corpus generated by machine translating the English AbstRCT corpus with DeepL and manually projecting the labels. We train T-Projection of the concatenation of the English Neoplasm train, dev and test split. We evaluate the systems by projecting the English Neoplasm train split.

## G Hardware used

We perform all our experiments using a single NVIDIA A100 GPU with 80GB memory. The machine used has two AMD EPYC 7513 32-Core Processors and 512GB of RAM.