# OpenReview forum: "T-Projection: High Quality Annotation Projection for Sequence Labeling Tasks"
_EMNLP/2023/Conference — EMNLP 2023 Findings_

### Official Review · Reviewer_nRSb · 2023-08-07

**Soundness:** 3

**Excitement:**

4: Strong: This paper deepens the understanding of some phenomenon or lowers the barriers to an existing research direction.

**Paper Topic And Main Contributions:**

This paper studies annotation projection --  a data argumentation technique to improve the performance of sequence labeling tasks in low-resource languages.  This task assumes the existence of labeled data in high-resource language and un-labeled in low-resource language. The goal is to annotate the data in low-resource languages automatically.

The authors propose a method called T-projection, which consists of two steps: (1) candidate generation by bream search using a fine-tuned mT5 model; (2) candidate selecting using translation probability.

The authors perform intrinsic evaluation by evaluating the accuracy of label projection. The proposed method outperforms the word alignment-based methods.

In the extrinsic evaluation, by incorporating labeled data generated by T-projection, the authors demonstrate a significant performance boost in three downstream sequence labeling tasks, including opinion target detection, NER, and argument mining, on five Indo-European languages and eight African languages supported by mT5.

**Questions For The Authors:**

I think this method heavily relies on the transferability of the mT5 model. Given a mT5 model trained in English, its inference on another language can have errors, for example, false positive and false negative.

I guess the false positive can be fixed by transition probability. Therefore the precision of this method should be high. But how about the recall? If some entities are missed, they can't be recovered, right? Is this a problem in practice?

**Reasons To Accept:**

1. The method is intuitive and makes sense. I like the design of symmetrized translation probability.
2. The experimental setup is reasonable and well executed. Strong baselines are included, and standard models are used.
3. I think the experimental results well support their claim.
4. The analysis is informative, showing why this method works.


**Reasons To Reject:**

N/A

**Reproducibility:**

4: Could mostly reproduce the results, but there may be some variation because of sample variance or minor variations in their interpretation of the protocol or method.

**Reviewer Confidence:**

4: Quite sure. I tried to check the important points carefully. It's unlikely, though conceivable, that I missed something that should affect my ratings.

---

> ### Author Rebuttal · Authors · 2023-08-28
>
> We greatly appreciate your comments and feedback. Regarding to your question about recall, it can be controlled by adjusting the number of generated candidates. Generating more candidates reduces the number of False Negatives. However, as the candidate set expands, it becomes noisier, leading to more False Positives, making the selection of the correct candidate increasingly challenging. In Section 5.2, we experimented with both generating just one candidate and considering every possible subsequence of the target sentence as candidates. We generated 100 candidates per entity for all our experiments. This decision was informed by a preliminary analysis of the performance at the candidate generation stage. In the appendix A, we conducted experiments with varying numbers of candidates, ranging from 5 to 100, and found that 10-25 candidates is the optimal range. We chose not to rerun all the experiments with the optimal number of candidates, as selecting a hyperparameter based on test set results is neither fair nor good research practice.

---

### Official Review · Reviewer_GnKV · 2023-08-10

**Soundness:** 4

**Excitement:**

3: Ambivalent: It has merits (e.g., it reports state-of-the-art results, the idea is nice), but there are key weaknesses (e.g., it describes incremental work), and it can significantly benefit from another round of revision. However, I won't object to accepting it if my co-reviewers champion it.

**Paper Topic And Main Contributions:**

This paper presents T-Projection, a new method for projecting sequence labeling annotations like named entities from one language to another for cross-lingual transfer learning. The key ideas are to use a text-to-text transformer to generate label projection candidates and then rank candidates using translation probabilities from a machine translation model.

**Questions For The Authors:**

a. Can you use llm(gpt3.5-burbo/text-davinci-003/gpt4) with simple end2end prompting or self-consistency prompting under few/zero-shot setting to complete the annotation projection task to prove that the proposed techniques is still a must in the era of llm.

**Reasons To Accept:**

* It proposes a. simple method, T-Projection, for cross-lingual annotation projection that achieves new state-of-the-art results. The technique is innovative in leveraging both multilingual pretrained language models and machine translation models.
* The gains over prior work are substantial. This is a large advance for a task where progress has been incremental.
* The method is evaluated extensively on 3 different sequence labeling tasks and 5-13 languages. The consistent sizable gains demonstrate the general applicability of the approach.
* The writing clearly explains the limitations of prior projection methods and how T-Projection addresses them. The overall paper is well-structured and easy to follow.

**Reasons To Reject:**

* The core idea lacks sufficient novelty. Using pretrained language models and machine translation models for cross-lingual transfer is naive and intuitive.
* The ablation studies are limited. More experiments could be done to provide insight into why this method works so well.
* The gains, while substantial, come from access to very large pretrained models. It is unclear if the techniques proposed are the key factors behind the improvement. If you substitute mt5 with gpt3.5-burbo/text-davinci-003/gpt4, maybe just simple end2end prompting can work better.

**Reproducibility:**

5: Could easily reproduce the results.

**Reviewer Confidence:**

4: Quite sure. I tried to check the important points carefully. It's unlikely, though conceivable, that I missed something that should affect my ratings.

---

> ### Author Rebuttal · Authors · 2023-08-28
>
> Thank you for your comments and feedback.
>
> As far as we are concerned, we don’t agree with the statement that our method being simple or intuitive involves a lack of novelty. The originality of our work resides in the way T-projection combines both text2text and translation models. Characterizing an approach as not novel because it uses pretrained language models and MT models would deem almost every single paper that gets published in NLP as "not novel".  We comprehensively reviewed  previous state-of-the-art on annotation projections and we present a new method which outperforms all of them. Therefore, we would appreciate references to previous work on annotation projection that justifies characterizing our work as “incremental” or “not novel”.
>
> OpenAI products are impressive. Unfortunately, as private products we don’t know much about the underlying models, how they were trained or which data was used. On top of that, this product is regularly updated and previous versions are not accessible, therefore, any experiment performed with OpenAI models would not be reproducible. Moreover, ChatGPT has been proven to massively underperform in named entity recognition and other information extraction / sequence labeling tasks (Lai et al.,2023, Tan et al., 2023). Besides, it does only support a few high-resource languages, so it is not suitable for projection annotations into low resource languages such as the African ones we use in our experiments.
>
> In contrast to ChatGPT, our models, data, and algorithms are publicly available, can be run locally, and our research can be reproduced by any other researcher.  Our system significantly outperforms current state-of-the-art systems and it is close to human performance.  The existence of an opaque model behind a paywall, so demanding that it requires the most powerful GPUs, doesn't diminish our contributions. In fact, we claim that achieving such an outstanding performance with a system that can run on consumer-grade hardware constitutes a remarkable strength of T-Projection.
>
> Regarding the ablation study, due to the 8-page limit, we were unable to include all the experiments in the paper. We selected those we deemed most relevant. In the appendix, you can find additional experiments that provide insight into why T-Projection performs so effectively. These include generating varying numbers of candidates, using different models and parameter sizes, adjusting hyper-parameters, and exploring alternative candidate selection algorithms. We would appreciate feedback on any other ablation experiments you believe might be relevant.
>
> ## References
>
> Lai, V. D., Ngo, N. T., Veyseh, A. P. B., Man, H., Dernoncourt, F., Bui, T., & Nguyen, T. H. (2023). Chatgpt beyond english: Towards a comprehensive evaluation of large language models in multilingual learning. arXiv preprint arXiv:2304.05613.
>
> Tan, Z., Huang, S., Jia, Z., Cai, J., Li, Y., Lu, W., ... & Jiang, Y. (2023). DAMO-NLP at SemEval-2023 Task 2: A Unified Retrieval-augmented System for Multilingual Named Entity Recognition. SemEval-2023

---

### Official Review · Reviewer_LiTa · 2023-08-18

**Soundness:** 3

**Excitement:**

3: Ambivalent: It has merits (e.g., it reports state-of-the-art results, the idea is nice), but there are key weaknesses (e.g., it describes incremental work), and it can significantly benefit from another round of revision. However, I won't object to accepting it if my co-reviewers champion it.

**Paper Topic And Main Contributions:**

This paper presents T-Projection, an innovative annotation projection methodology that capitalizes on the capabilities of extensive multilingual pretrained text2text language models in conjunction with leading-edge machine translation systems. Through comprehensive experimental evaluations encompassing both intrinsic and extrinsic tasks across five Indo-European languages and eight African languages, our findings underscore the superiority of T-Projection over established annotation projection techniques. Consequently, T-Projection establishes a new benchmark in the realm of annotation projection tasks.

**Questions For The Authors:**

1) As observed in Table 1, both Giza++ and FastAlign, given their antiquity, might seem unexpected choices for inclusion. Nevertheless, it is intriguing to note their commendable F1 score performance on the AM task. Additionally, there's a notable decline in performance for Span Translation on the AM task. It would be beneficial for the authors to delve into these phenomena for clearer understanding.

2) Upon examination, it appears that T-Projection primarily employs pre-existing translation models. Consequently, one could postulate that enhancements in the underlying translation model could potentially boost T-Projection's performance, correct?

3) It would be instructive for the author to provide a detailed analysis concerning name translation. Specifically, delineating which segments of a name undergo translation, transliteration, or remain unaltered, stratified by each language.

**Reasons To Accept:**

This study offers detailed experimental evaluations, covering both intrinsic and extrinsic tasks, across five Indo-European languages and eight African languages. Furthermore, a thorough analysis elucidates the pivotal roles of both the generation candidate and the candidate selection phases in determining the ultimate performance.

**Reasons To Reject:**

The proposed method lacks originality, given that T-Projection predominantly relies on machine translation systems, which historically grapple with the intricacies of translating names. Arguably, the paper would be more impactful if it solely centered on Named Entities. This specificity is justified by the inherent complexities of name translations/transliterations across languages, setting them apart from aspects like Opinion Target or Argument Mining.

**Reproducibility:**

5: Could easily reproduce the results.

**Reviewer Confidence:**

4: Quite sure. I tried to check the important points carefully. It's unlikely, though conceivable, that I missed something that should affect my ratings.

---

> ### Author Rebuttal · Authors · 2023-08-28
>
> Thank you for your comments and feedback.
>
> Firstly, we made a thorough revision on  previous state-of-the-art on annotation projections, and no other prior work employs text2text models to generate annotation projection candidates and uses translation models to rank those candidates. Furthermore, T-Projection outperforms any other previously proposed method both in intrinsic and extrinsic evaluations by a wide margin. If lack of originality is going to be claimed, it would be appreciated for the reviewer to provide references to previous works to support the claim.
>
> Secondly, we would like to clarify our use of translation models in T-Projection. Translation models are never used to translate spans of text in T-Projection, they are solely used to rank the projection candidates generated by an mT5 model. During inference, T-Projection uses source sentence labels’ and the unlabeled target sentence as input. Any generated candidate by mT5 that is not a subsequence of the target sentence is discarded (Section 3.2). Therefore, name translation/transliteration does not take place at any step in T-Projection; we use words as they appear in the target sentence. As empirically demonstrated in Section 5.2, this novel combination of using text-to-text language models to generate candidates and  using translation models to select the best candidate is the source of T-Projection's performance. How the target sentence is generated and the potential mistranslations that may arise if they are obtained using machine translation systems are beyond the scope of our paper.
>
> Enhancements to the underlying translation model can improve T-Projection performance, since more accurate translation probabilities would result in a better selection of candidates (section 3.2). In Section 5.2, we compute the upper bound scores of always choosing the correct candidate. This enables us to analyze the pre-trained translation model quality that we use in the candidate selection step.
>
> It's true that Giza++ and FastAlign are both based on the IBM alignment models from the '90s. However, these models have been the state-of-the-art in many word alignment and annotation projection algorithms and can be considered as baseline systems. While transformer-based word-alignment systems have surpassed these statistical models, they are extremely cost-effective to train and use for inference, and their performance remains competitive. Given their exceptional cost/performance ratio and for historical reasons, we find it valuable to include them as baselines.
>
> The Argument Mining (AM) task differs from other tasks in the length of the annotated spans. In this task, the annotated spans are notably long. The baselines were computed using the algorithm from Garcia-Ferrero et al., 2022, to generate annotation projections from word-alignment graphs. This algorithm rectifies “gaps” in the word-alignment graph (“If a labeled span in the source sentence is split into two or more parts in the target sentence, we fill the gap, considering the unlabeled words between the parts as labeled words of the same class if the gap is one word”). With longer sequences, the algorithm becomes more robust, and the quality of the word alignments becomes less critical as long as the first and last words of the projected spans are accurate. As for AWESOME, it underperforms in the AM task because it consistently misaligns articles at the beginning of sentences; a detailed study on this was conducted by Yeginbergenova et al., 2023. Regarding the Span Translation baseline,  the performance of this baseline degrades with long annotated spans, as the likelihood of producing a translation that is a subsequence of the target sentence decreases with the number of tokens in the labeled span. The Span Translation baseline was improved by Chen et al,. 2023. We also compare T-projection with their work (EasyProject), although as explained in section Section 4.2, it was only possible for us to compare T-Projection with EasyProject in the extrinsic evaluation.
>
> ## References
> García-Ferrero, I., Agerri, R., & Rigau, G. (2022, December). Model and Data Transfer for Cross-Lingual Sequence Labelling in Zero-Resource Settings. In Findings of the Association for Computational Linguistics: EMNLP 2022 (pp. 6403-6416).
>
> Yeginbergenova, A., & Agerri, R. (2023). Cross-lingual Argument Mining in the Medical Domain. arXiv preprint arXiv:2301.10527.
>
> Chen, Y., Jiang, C., Ritter, A., & Xu, W. (2023). Frustratingly easy label projection for cross-lingual transfer. In Findings of the Association for Computational Linguistics: ACL 2023

---

### Meta-Review · Area_Chair_EjuQ · 2023-09-18

**Recommendation:** 3

**Metareview:**

The article presents a new method for projecting annotations, which improves on the performance of the state of the art. The reviewers suggested a number of experiments to better understand and analyse the results, which I think are worthwhile, which is why I think this article is only “borderline”.

---

### Decision · Program_Chairs · 2023-10-07

**Decision:**

Accept-Findings

**Comment:**

The article presents a new method for projecting annotations, which improves on the performance of the state of the art. The reviewers suggested a number of experiments to better understand and analyse the results, which I think are worthwhile, which is why I think this article is only “borderline”.